# SH003 as a Redox-Immune Modulating Phytomedicine: A Ferroptosis Induction, Exosomal Crosstalk, and Translational Oncology Perspective

**DOI:** 10.3390/cancers17213519

**Published:** 2025-10-31

**Authors:** Moon Nyeo Park, Md. Maharub Hossain Fahim, Han Na Kang, Hanul Bae, Amama Rani, Fahrul Nurkolis, Trina E. Tallei, Seong-Gyu Ko, Bonglee Kim

**Affiliations:** 1Department of Pathology, College of Korean Medicine, Kyung Hee University, Seoul 02447, Republic of Korea; mnpark@khu.ac.kr (M.N.P.); maharubhossain@khu.ac.kr (M.M.H.F.); elegantsky@khu.ac.kr (H.B.); amama.rani@khu.ac.kr (A.R.); 2KM Convergence Research Division, Korea Institute of Oriental Medicine, Daejeon 34054, Republic of Korea; khn3110@kiom.re.kr; 3Department of Zoology, University of Azad Jammu and Kashmir, Muzaffarabad 13100, Pakistan; 4Master of Basic Medical Science, Faculty of Medicine, University Airlangga, Surabaya 60115, Indonesia; fahrul.nurkolis-2024@fk.unair.ac.id; 5Faculty of Science and Technology, State Islamic University of Sunan Kalijaga (UIN Sunan Kalijaga), Yogyakarta 55281, Indonesia; 6Department of Biology, Faculty of Mathematics and Natural Sciences, University Sam Ratulangi, Manado 95115, Indonesia; trina_tallei@unsrat.ac.id; 7Department of Biology, Faculty of Medicine, University Sam Ratulangi, Manado 95115, Indonesia; 8Korean Medicine-Based Drug Repositioning Cancer Research Center, College of Korean Medicine, Kyung Hee University, Seoul 05253, Republic of Korea

**Keywords:** phytomedicine, redox signaling, ferroptosis, NRF2–KEAP1, STAT3/PD-L1, exosomal microRNAs, precision oncology

## Abstract

**Simple Summary:**

Cancer cells frequently evade cell death by suppressing ferroptosis and remodeling the immune microenvironment. SH003, a GMP-standardized herbal formulation, has shown promising preclinical and early clinical evidence of safety and efficacy. This review highlights how SH003 regulates redox signaling, induces ferroptotic vulnerability, and enhances antitumor immunity through STAT3/PD-L1 inhibition and macrophage/T cell activation. These network-level effects suggest SH003 as a representative phytomedicine that bridges traditional herbal therapy with modern precision oncology.

**Abstract:**

Redox dysregulation, ferroptosis evasion, and immune suppression are major barriers in cancer therapy. SH003, a multi-herbal formulation standardized under GMP conditions and evaluated in early-phase clinical studies (NCT03081819; KCT0004770), demonstrated a favorable safety profile supporting its translational potential. Preclinical studies reveal that SH003 disrupts mitochondrial homeostasis, triggers endoplasmic reticulum stress apoptosis, and sensitizes resistant tumors to ferroptosis via suppression of the SLC7A11–GPX4 axis and NRF2 destabilization. In parallel, SH003 remodels tumor immunity by attenuating STAT3-driven PD-L1 signaling, promoting macrophage repolarization, and enhancing cytotoxic lymphocyte activity. Exosome-associated microRNAs further suggest SH003’s role in redox–immune communication, although functional validation is pending. Collectively, SH003 represents a clinically tested phytomedicine that integrates ferroptosis induction with immune modulation, offering a biomarker-informed approach to precision oncology.

## 1. Introduction

Reactive oxygen species (ROS) function as pivotal regulators of cellular proliferation, survival programs, and immune activity. Malignant cells exploit ROS-dependent signaling to drive oncogenic transformation, sustain metabolic reprogramming, and adapt to therapeutic stress—a phenomenon often described as ROS addiction [1,2]. This dependency not only accelerates tumor progression but also creates exploitable liabilities. Attempts to therapeutically target ROS imbalance using conventional modulators have faced major obstacles, including insufficient pathway specificity, systemic toxicity, and the rapid development of adaptive resistance. Moreover, single-pathway inhibitors rarely succeed against the remarkable plasticity of redox-addicted malignancies such as triple-negative breast cancer (TNBC) and non-small cell lung cancer (NSCLC).

SH003, a GMP-standardized and authenticated multi-herbal formulation derived from *Astragalus membranaceus*, *Angelica gigas*, and *Trichosanthes kirilowii*, has emerged as a representative example of next-generation phytomedicine [3,4]. Distinct from traditional small-molecule agents, SH003 modulates interconnected redox-sensitive processes—spanning mitochondrial ROS imbalance, endoplasmic reticulum (ER) stress, ferroptosis sensitization, and immune–exosome reprogramming—thereby functioning as a broad-spectrum platform for biomarker-informed interventions. In preclinical models, SH003 shows selective cytotoxicity in TNBC and NSCLC, and early-phase clinical evaluations (NCT03081819; KCT0004770) support a favorable safety profile, underscoring translational feasibility.

From a pharmacological standpoint, SH003 is among the earliest multi-herbal formulations formally evaluated in oncology, and its capacity to disrupt adaptive redox networks while remodeling immune dynamics highlights the potential of complex phytomedicines to advance precision oncology paradigms [5,6].

In first-in-human and multicenter Phase I studies, SH003 reached a maximum tolerated dose (MTD) of 4800 mg/day as monotherapy and was combinable with docetaxel without SH003-attributed dose-limiting toxicities; an expanded Phase I established safety up to 9600 mg/day, collectively indicating a favorable human safety window [7,8,9]. These trials also documented manufacturing traceability and rigorous authentication/quality-control procedures (e.g., marker-compound validation for decursin and formononetin), supporting reproducibility and clinical traceability expected for herbal formulations [7,8,9]. SH003 is composed of three major bioactive constituents: formononetin from *Astragalus membranaceus*, decursin from *Angelica gigas*, and cucurbitacin D from *Trichosanthes kirilowii*. Pharmacokinetic studies have shown that formononetin exhibits oral bioavailability of approximately 20–22% and a half-life of 2–4 h in rodents, primarily absorbed via passive diffusion and subject to rapid Phase II metabolism through UGT and CYP enzymes [10,11]. Decursin undergoes extensive first-pass hydrolysis to decursinol and hepatic oxidation, while cucurbitacin D displays high lipophilicity and CYP3A-dependent clearance [12]. These characteristics suggest that SH003’s constituents achieve multi-target systemic exposure with limited toxicity and predictable metabolic profiles.

### Review Methodology

This article represents a comprehensive narrative review integrating mechanistic, preclinical, and early clinical evidence on SH003. Literature was collected from PubMed, Scopus, and Web of Science databases (2010–2025) using the Boolean combination of keywords ‘SH003’ AND (‘redox’ OR ‘ROS’ OR ‘ferroptosis’ OR ‘NRF2’ OR ‘STAT3’ OR ‘exosome’ OR ‘immune modulation’). Inclusion criteria comprised in vitro, in vivo, and clinical studies related to SH003 or its constituent compounds. Reviews, abstracts without mechanistic data, and unrelated natural products were excluded. Two authors independently screened the records to minimize bias.

## 2. Mechanistic Basis of SH003 in Redox Biology

### 2.1. Exosome-Immune Crosstalk and NRF2 Modulation

SH003-induced oxidative stress extends beyond intracellular compartments [13], influencing both exosomal cargo composition and immune regulatory pathways [14,15,16,17,18,19]. Exosomes carrying redox-sensitive miRNAs (miR-200c, miR-21, miR-210, miR-96) have been reported to modulate ROS buffering [20,21,22,23], ferroptosis sensitivity, and PD-L1 expression, thereby reshaping tumor–immune interactions [24]. In triple-negative breast cancer, tumor- and stroma-derived vesicles can activate the HDAC6/STAT3/PD-L1 signaling cascade, facilitating immune evasion [25]. In NSCLC models, SH003 ± docetaxel exerts synergistic antitumor effects via EGFR/STAT3 blockade; while exosome-based metabolomic signatures are best positioned as exploratory predictive biomarkers rather than proximal effectors of response, pending prospective immune-functional validation [26,27].

At the phytochemical level, individual SH003 constituents provide mechanistic plausibility: formononetin downregulates STAT3/PD-L1 signaling, baicalein promotes macrophage M1 polarization, and luteolin enhances CD8^+^ T-cell activity [6]. Current evidence, however, is largely inferred from constituent-based studies and awaits validation in SH003-specific systems. Beyond vesicle-mediated signaling, SH003 may also destabilize NRF2-driven antioxidant adaptation by modulating GSK3β activity, thereby providing KEAP1-independent regulation. This dual control suggests that SH003 could impair redox resilience in NRF2-hyperactivated tumors, offering opportunities for biomarker-guided patient stratification [28,29]. Collectively, SH003 exemplifies a phytomedicine capable of linking exosomal signaling, immune remodeling, and NRF2 regulation, underscoring its potential as a prototype redox–immune modulator in precision oncology (Figure 1).

### 2.2. Ferroptosis Induction

Ferroptosis is an iron-dependent form of regulated cell death, characterized morphologically by shrunken mitochondria and biochemically by glutathione (GSH) depletion, glutathione peroxidase 4 (GPX4) inactivation, and accumulation of lipid peroxides (LPOs) [30]. Although ferroptosis provides an attractive therapeutic target in cancer and inflammatory disorders, translation into the clinic remains challenging owing to tumor redox plasticity, the complexity of iron metabolism, and potential risks of off-target oxidative injury [30,31]. Experimental evidence demonstrates that genetic ablation of GPX4 results in embryonic lethality and triggers tissue-specific ferroptosis [30]. Likewise, disruption of glutathione S-transferase alpha 4 (GSTA4) enhances ferroptotic sensitivity in macrophages and prevents microbiota-driven colorectal tumorigenesis, highlighting ferroptosis as both a cytotoxic pathway and a regulator of immune–tumor interactions [31]. SH003 enforces ferroptotic vulnerability through multiple mechanisms: cucurbitacin-mediated inhibition of the SLC7A11–GPX4 axis, flavonoid-associated perturbation of iron handling [32,33,34], and NRF2 suppression by formononetin [35]. In contrast to single-pathway inducers, SH003 may amplify via multi-node interference, sensitizing ferroptosis-resistant tumor types such as triple-negative breast cancer (TNBC) and epidermal growth factor receptor (EGFR)-mutant NSCLC [36,37,38]. From a pharmacological perspective, ferroptosis induction by SH003 provides a rational approach to bypass therapeutic resistance and may serve as a foundation for biomarker-guided treatment strategies.

### 2.3. NRF2-KEAP1 and Redox Adaptation

The NRF2–KEAP1 signaling axis coordinates cellular antioxidant defenses and plays a central role in therapy resistance [39]. Numerous phytochemicals have been shown to modulate NRF2 activity in divergent ways [40], either sensitizing malignant cells to oxidative stress or attenuating adaptive survival mechanisms [41,42]. Through indirect, multi-target interference [43], SH003 redefines NRF2 not simply as a protective regulator but as a pharmacological vulnerability, effectively lowering cellular redox thresholds and thereby augmenting both ferroptotic and apoptotic susceptibility [35,41]. Given the high prevalence of NRF2 hyperactivation in KEAP1-mutant NSCLC—and its established links to immune escape and drug resistance—KEAP1/NRF2 genetic background emerges as a clinically relevant biomarker to stratify patients for SH003-based therapeutic strategies [44,45,46,47]. Prospective SH003 trials should pre-specify KEAP1/NFE2L2 genotype as a randomization/stratification covariate and include correlative endpoints—such as an NRF2 target-gene signature and serum SQSTM1 kinetics—to test whether NRF2-hyperactivated tumors derive preferential benefit from SH003-based regimens [48,49].

### 2.4. ER Stress and Apoptosis

The unfolded protein response (UPR) enables cells to adapt to oxidative disturbances; however, sustained or excessive activation redirects this signaling toward apoptosis [50]. SH003 constituents, including decursin [51,52] and hispidulin, promote this pro-apoptotic branch through the PKR-like ER kinase (PERK)–eukaryotic initiation factor 2α (eIF2α)–activating transcription factor 4 (ATF4)–C/EBP homologous protein (CHOP) cascade, thereby converting an adaptive UPR into a cytotoxic outcome. Functional crosstalk with mitochondrial depolarization and caspase activation further highlights SH003’s ability to exploit ER–mitochondrial stress coupling in redox-vulnerable malignancies.

CHOP activation is widely recognized as a hallmark of ER stress–driven apoptosis [53]. Experimental evidence from preclinical models—including sepsis-induced muscle wasting and renal injury—shows that CHOP induction accelerates maladaptive protein degradation and cell death, whereas CHOP deletion or pathway inhibition alleviates these pathological consequences [54,55]. Clinically, CHOP has been investigated as a biomarker of ER stress in metabolic conditions such as gestational diabetes mellitus [56]. Taken together, these findings suggest that CHOP upregulation could serve as a dynamic and clinically relevant biomarker of ER stress engagement in SH003-treated cancers, warranting further validation in patient-derived cohorts.

### 2.5. Mitochondrial ROS Disruption

Mitochondria function both as major sources of reactive oxygen species (ROS) and as central regulators of intrinsic apoptosis [57]. SH003-derived phytochemicals, including luteolin and baicalein, perturb mitochondrial homeostasis by suppressing fusion proteins, inducing Bcl-2–associated X protein (BAX)/BAK oligomerization, promoting cytochrome c release, and activating caspases [58]. Through selective disruption of mitochondria with impaired antioxidant defenses, SH003 acts as a targeted mitochondrial redox disruptor. From a pharmacodynamic standpoint, components of the mitochondrial thioredoxin system (Trx/TrxR) constitute core antioxidant circuitry and plausible readouts of oxidative-stress engagement; notably, Trx/TrxR operate alongside SOD/GPX defenses in mitochondria and are functionally coupled to SIRT3-dependent redox control [59,60]. Markers of mitochondrial dysfunction—particularly changes in thioredoxin 2 (TRX2)—have been identified as sensitive indicators of oxidative imbalance. Because TRX2 is critical for mitochondrial ROS detoxification and regulation of apoptosis, its modulation offers both mechanistic insight and a potential pharmacodynamic endpoint for clinical monitoring. Inhibition of the thioredoxin system has been shown to drive ROS accumulation, lipid peroxidation, and either ferroptotic or apoptotic cell death, while early-phase clinical evaluation of TRX inhibitors such as PX-12 reported acceptable safety and tolerability in patients with advanced solid tumors [61]. Collectively, these findings support the translational utility of TRX2 modulation as a biomarker to guide patient stratification and therapeutic response assessment in SH003-based oncology strategies.

### 2.6. ROS-Generation Properties of SH003 Constituents

The pro-oxidant activity of SH003 arises from the inherent redox characteristics of its phytochemical components. Luteolin acts as a strong ROS generator through catechol-driven redox cycling [62], whereas baicalein produces more moderate ROS elevation by disrupting mitochondrial stability and enhancing endoplasmic reticulum (ER) stress. Even constituents present at lower abundance, such as cucurbitacin D, contribute synergistically by weakening intrinsic antioxidant defenses. Together, these phytochemicals enable SH003 to operate as a network-level modulator of ROS rather than a simple additive effect of single compounds [63,64]. This integrated mode of redox regulation represents a pharmacological mechanism distinct from conventional single-target approaches and highlights the translational potential of multi-component herbal formulations in oncology [65,66]. A comparative summary of constituent-specific functions—including chelation capacity, biomarker relevance, ROS modulation, and anticancer mechanisms—is provided in Table 1.

### 2.7. Integrated Perspective

From a systems biology standpoint, SH003 is best conceptualized not as a simple aggregation of individual phytochemicals but as a multi-dimensional modulator of redox networks. By simultaneously engaging apoptotic, ferroptotic, and autophagic programs, SH003 illustrates a prototypical approach for counteracting tumor plasticity and drug resistance. Pharmacologically, this integrated mode of action positions SH003 as a systems-level therapeutic strategy, capable of targeting diverse redox-dependent vulnerabilities in cancer.

### 2.8. Biomarkers and Translational Perspectives

Mechanism-linked biomarker development is fundamental to realizing the translational potential of SH003 [78]. Low GPX4 expression identifies tumors vulnerable to ferroptosis [79], while NRF2 hyperactivation may serve as a predictor of benefit from redox-targeting strategies [80]. CHOP induction reflects engagement of ER stress pathways, and alterations in thioredoxin 2 (TRX2) indicate mitochondrial redox imbalance [81]. While CHOP has shown clinical associations—ranging from serum detectability in critical-illness cohorts to predictive value for chemotherapy outcomes in breast cancer—its oncology-specific use as a pharmacodynamic biomarker for SH003 remains to be prospectively validated, and would benefit from integration with ER-stress readouts and adaptive UPR dynamics in early-phase trials [82,83,84]. As a preclinical corollary, the PERK–ATF4–CHOP axis functions as a gatekeeper of drug/TNFα-induced hepatocyte apoptosis in hepatotoxicity models, reinforcing CHOP’s utility as an ER-stress pharmacodynamic readout [85]. Collectively, these biomarkers offer a framework for patient stratification and dynamic pharmacodynamic monitoring in SH003-based oncology regimens [86]. Clinically, Phase I studies demonstrated safety at doses up to 9600 mg/day without dose-limiting toxicities, whereas combination trials with docetaxel established a maximum tolerated dose of 4800 mg/day. Incorporating biomarker analysis into such trials could support precision-guided application and distinguish SH003 from conventional phytomedicines [7]. An integrated model of SH003-mediated redox modulation—spanning ferroptosis, autophagy, and immune remodeling—is depicted in Figure 2, while Table 2 summarize biomarker associations and constituent-specific mechanistic evidence.

## 3. Discussion and Translational Perspectives

SH003 imposes ferroptotic stress through the combined actions of cucurbitacin D, luteolin, and baicalein, which synergistically enhance lipid peroxidation and suppress GPX4 activity [90]. This polypharmacological pressure differentiates SH003 from conventional single-pathway inducers and offers a strategy to overcome ferroptosis resistance in redox-adapted tumors. Ferroptosis, an iron-dependent form of regulated cell death characterized by GSH depletion, GPX4 inactivation, and lipid peroxide accumulation, has emerged as a promising anticancer strategy. Yet, clinical translation has been hindered by tumor redox plasticity, iron homeostasis complexity, and concerns about off-target oxidative injury [30,31].

Mechanistic studies illustrate this duality: genetic GPX4 ablation results in embryonic lethality with tissue-specific ferroptosis [30], whereas inhibition of glutathione S-transferase alpha 4 sensitizes macrophages and prevents colorectal tumorigenesis [31]. These findings underscore ferroptosis as both a cell-death pathway and an immunoregulatory mechanism. Recent advances reveal that SH003 constituents intricately modulate iron homeostasis, thereby influencing ferroptotic vulnerability in tumor cells. Luteolin forms stable Fe^2+^ and Fe^3+^ chelates through its catechol (3′,4′-dihydroxy) and 5-hydroxy-4-keto coordination sites, which effectively sequester labile iron and suppress Fenton reaction–driven hydroxyl radical generation [91]. This chelating property contributes to reduced lipid peroxidation and preserves membrane integrity under oxidative stress. Baicalein, another major SH003 flavone, exhibits dual functionality: it acts as a potent iron chelator via its 6-,7-dihydroxy moieties while also stabilizing mitochondrial iron pools, thereby preventing Fe^2+^-induced ROS bursts and GPX4 inactivation. In contrast, cucurbitacin D has been reported to enhance ferritinophagy through NCOA4 activation, liberating stored ferritin-bound Fe^2+^ and amplifying lipid peroxidation, thus promoting ferroptotic stress in resistant cancer cells. This balanced regulation of iron sequestration (via luteolin and baicalein) and iron release (via cucurbitacin D) suggests that SH003 may achieve a homeostatic ferroptotic threshold—sensitizing tumors to redox imbalance while minimizing systemic oxidative toxicity. Mechanistically, ferroptosis is driven by iron-dependent lipid peroxidation and GPX4 depletion, but recent studies highlight that perturbation of the labile iron pool, ferritinophagy, and mitochondrial Fe^2+^ overload are decisive in dictating ferroptotic sensitivity [92]. Therefore, integrating iron-handling biomarkers such as ferritin heavy chain (FTH1), ferritin light chain (FTL), transferrin receptor 1 (TFRC), and ferroportin (SLC40A1) into SH003-based studies could clarify its dual iron-chelating and ferroptosis-sensitizing effects, offering a more structured and translationally relevant mechanistic backbone. In this context, SH003’s advancement to Phase I trials (NCT03081819, KCT0004770) underscores its rare translational trajectory as a GMP-standardized multi-herbal formulation [93]. By bridging phytochemistry, pharmacology, and oncology—exemplified by recent studies on the ROS–miRNA–exosome axis [16]—SH003 provides a framework for integrating complex phytomedicines into precision cancer therapy.

Mechanistic alignment with tumor vulnerabilities—including redox addiction, ferroptosis resistance, and immune evasion—supports its relevance for triple-negative breast cancer (TNBC) and EGFR-mutant NSCLC [94,95]. Biomarker-defined subgroups, such as tumors with low GPX4, NRF2 hyperactivation [96], or elevated CHOP/TRX2 [97], may be particularly responsive to SH003-driven stress responses [47,98]. Combination strategies further expand clinical potential: SH003 can enhance PD-L1 suppression and restore T-cell cytotoxicity when combined with Immune Checkpoint Inhibitors (ICIs) [6,99], amplify lipid peroxidation when paired with ferroptosis sensitizers [24,100,101], and has already shown synergy with docetaxel in overcoming resistance while mitigating systemic toxicity [102,103,104]. Although exosome remodeling by SH003 has been supported by metabolomics, its downstream immune consequences warrant validation [105]. In NSCLC models, SH003 ± docetaxel suppresses the EGFR–JAK–STAT3 axis and yields synergistic antitumor activity; accordingly, exosome-based metabolomic signatures in this setting are best positioned as exploratory predictive biomarkers rather than mechanistic effectors, pending prospective immune-functional validation [6,99,102,103,104,105]. Recent mechanistic studies have demonstrated that SH003 and its constituents inhibit the JAK/STAT pathway through both direct and indirect mechanisms. Formononetin and cucurbitacin D suppress JAK2 and STAT3 Tyr705 phosphorylation, hinder STAT3 dimerization and nuclear translocation, and consequently decrease PD-L1 expression and downstream oncogenic targets such as c-Myc and Bcl-xL [106,107]. In parallel, SH003 mitigates ROS-NF-κB–IL-6 signaling, thereby reducing cytokine-driven STAT3 activation and remodeling exosomal cargo, including PD-L1 and immunoregulatory miRNAs (miR-21, miR-155) [107,108]. This dual-level inhibition links redox regulation to immune reactivation, enhancing M1 macrophage polarization and CD8^+^ T-cell activity while counteracting NRF2-dependent antioxidant and immunosuppressive pathways [107].

In parallel, redox-sensitive miRNAs (miR-200c, miR-96, miR-21, miR-210) are recognized as biomarkers of prognosis, resistance, and immune modulation across cancers [21,31,109,110,111], and their integration into SH003-based interventions could enable biomarker-guided oncology [112].

Beyond exosomal signaling, SH003 may regulate NRF2 stability via GSK3β, providing a KEAP1-independent layer of redox control. This dual regulation highlights SH003’s capacity to destabilize redox adaptation in NRF2-hyperactivated tumors, reinforcing the rationale for biomarker-driven stratification [28,29]. Embedding assays such as GPX4, NRF2, CHOP, and TRX2 into clinical trials will be critical for distinguishing SH003 from conventional phytomedicines.

Most breast cancer trials focus on supportive endpoints—radiodermatitis prevention, perioperative anxiety relief, or menopausal symptom control [113,114]. In contrast, lung cancer studies increasingly explore disease-modifying effects [115,116], including tumor suppression, resistance reversal, and immune modulation [117,118]. This contrast underscores SH003’s unique positioning: unlike supportive agents, SH003 directly enforces ferroptotic pressure, destabilizes NRF2-driven adaptation, and remodels immune–exosome networks.

Conventional chemotherapy (taxanes, platinum-based agents), EGFR-tyrosine kinase inhibitors (TKIs) [116], and ICIs have improved outcomes in breast [119] and lung cancer but remain constrained by toxicities [120], acquired resistance, and immune-related adverse events (irAEs) [115]. Recent herbal interventions have sought to mitigate these issues—for example, alleviating radiodermatitis in breast cancer [114,121], reducing chemotherapy-induced gastrointestinal toxicities in NSCLC [117,118], or improving QoL with immune modulation. Yet, most remain supportive rather than disease-modifying. In contrast, SH003 integrates cytotoxic, ferroptotic, and immunoregulatory mechanisms with favorable safety in Phase I trials, positioning it as a forward-looking phytomedicine capable of synergistic integration into precision oncology. This positioning is further contextualized by Table 3, where most natural product interventions remain supportive rather than disease-modifying. By contrast, SH003—mechanistically anchored and clinically advancing—illustrates how phytomedicine can evolve into a truly disease-modifying oncology therapeutic.

### Limitations and Future Directions

SH003-induced exosome remodeling and miRNA-mediated immune regulation are based on indirect or constituent-level studies, necessitating SH003-specific functional validation. In addition, critical aspects of iron–sulfur cluster balance, cancer stem cell metabolic dependencies, and nutrient-stress-conditioned ferroptotic switching remain unresolved. Clinically, available data are restricted to Phase I trials, which confirm safety but do not establish efficacy. Finally, as a multi-component phytomedicine, the precise contribution of individual constituents to overall therapeutic effects have yet to be delineated. Addressing these limitations will require multi-omics integration, spatial transcriptomics, and biomarker-embedded clinical studies to ensure SH003 advances from a supportive adjunct to a precision-guided therapeutic. To operationalize this agenda, we propose an exploratory clinical framework emphasizing KEAP1/NFE2L2-based stratification, NRF2 target-gene signatures, GPX4/PD-L1 status, and correlative pharmacodynamic readouts (including CHOP and the thioredoxin/thioredoxin-reductase system such as TRX2), with exosomal miRNA and metabolomic signatures positioned as exploratory predictive biomarkers, as summarized in Table 4. Future research should exploit the convergence of ferroptosis and immunotherapy. Nanoparticle-based delivery systems could optimize SH003 dosing and tumor targeting, while combinatorial approaches with CAR-T cells, PD-1/PD-L1 blockade, or STING agonists may potentiate cytotoxic T-cell activity and overcome microenvironmental resistance. Such multimodal strategies can advance SH003 from an adjunctive formulation toward a precision-guided therapeutic platform.

## 4. Conclusions

SH003 exemplifies how a GMP-standardized multi-herbal formulation can progress from mechanistic discovery to clinical translation. By modulating ferroptosis, NRF2 adaptation, and line immune remodeling through STAT3/PD-L1 suppression, SH003 functions as a multi-node redox–immune modulator. These convergent mechanisms provide a foundation for biomarker-informed patient stratification, particularly in GPX4-low or NRF2-hyperactivated tumors. Future clinical studies integrating predictive biomarkers such as GPX4, NRF2, CHOP, TRX2, and exosomal miRNAs will be crucial to validate SH003 as a prototype phytomedicine for precision oncology.

## Figures and Tables

**Figure 1 cancers-17-03519-f001:**
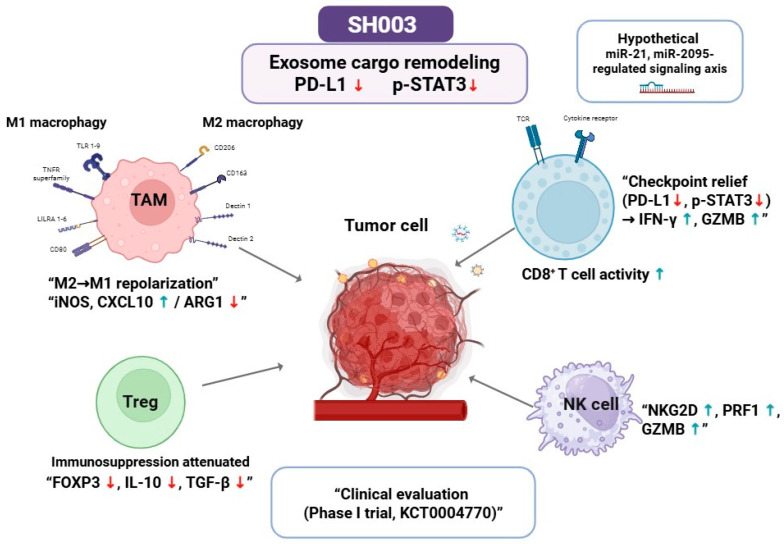
Proposed and experimentally supported mechanisms of SH003 in regulating ferroptosis, redox signaling, and immune remodeling. In early-phase clinical testing (KCT0004770), SH003 administered with docetaxel demonstrated a favorable safety profile and potential therapeutic synergy, underscoring translational feasibility. Mechanistically, SH003 is proposed to remodel tumor-derived exosome cargo, thereby reprogramming the tumor–immune interface. Major features include: (i) repolarization of macrophages toward an M1 phenotype (iNOS, ARG1, CXCL10 upregulation); (ii) attenuation of regulatory T-cell–mediated suppression through reduced FOXP3, IL-10, and TGF-β; (iii) restoration of CD8^+^ T-cell effector function via STAT3 inhibition and PD-L1 downregulation, accompanied by enhanced CD8^+^/NK infiltration (in synergy with docetaxel); and (iv) activation of NK cells with increased NKG2D, PRF1, and GZMB expression. Representative exosomal miRNAs (miR-200c, miR-21) are illustrated as cargo mediators linking redox signaling to immune remodeling. Dashed outlines indicate hypothesized or incompletely validated pathways. Overall, SH003 is depicted as a model phytomedicine that integrates exosomal regulation and immune modulation within a biomarker-informed therapeutic framework. Abbreviations: serine/threonine kinase (Akt); arginase (ARG); cluster of differentiation 8 positive T lymphocyte (CD8^+^ T cell); C-X-C motif chemokine ligand 10 (CXCL10); forkhead box P3 (FOXP3); granzyme B (GZMB); inducible nitric oxide synthase (iNOS); interferon-gamma (IFN-γ); classically activated macrophage (pro-inflammatory) (M1 macrophage); alternatively activated macrophage (anti-inflammatory) (M2 macrophage); natural killer cell (NK cell); natural killer group 2 member D (NKG2D); perforin 1 (PRF1); regulatory T cell (Treg); tumor-associated macrophage (TAM); T-cell receptor (TCR); transforming growth factor-beta (TGF-β). Red arrows indicate downregulation, and blue arrows indicate upregulation of the corresponding proteins or pathways. Created in BioRender. Kim, B. (2025) https://BioRender.com/6e3e802 (accessed on 28 October 2025).

**Figure 2 cancers-17-03519-f002:**
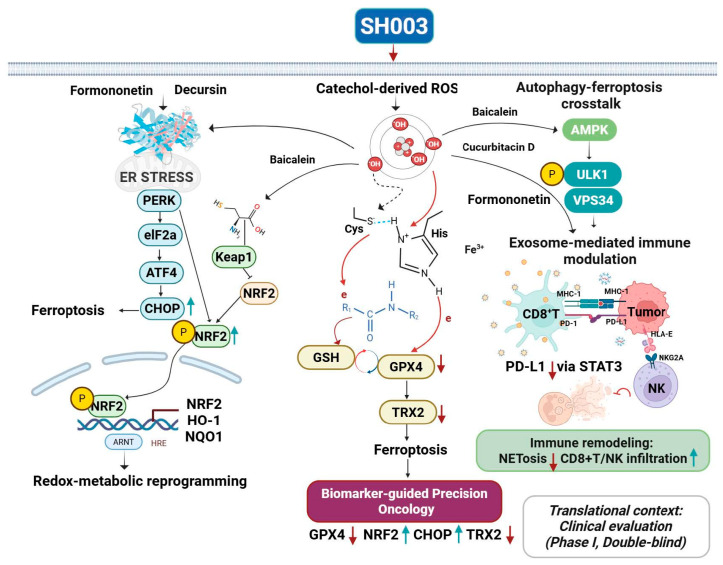
Integrated redox–ferroptosis–immune modulation by SH003. SH003 phytochemicals (decursin, formononetin, baicalein, cucurbitacin D, luteolin) target interconnected pathways including ER stress, NRF2–KEAP1 antioxidant signaling, ferroptosis, and autophagy. These effects converge on immune remodeling via STAT3/PD-L1 suppression and enhanced CD8^+^ T/NK infiltration. Representative biomarker alterations (GPX4, NRF2, CHOP, TRX2) highlight its potential for patient stratification. Translationally, SH003 has progressed to Phase I clinical trials, establishing safety and feasibility. Abbreviations: neutrophil extracellular trap–mediated cell death (NETosis); Sequestosome 1 (p62/SQSTM1); Thioredoxin 2 (TRX2); Unc-51 like autophagy activating kinase 1 (ULK1); Vacuolar protein sorting 34 (VPS34). Red arrows indicate downregulation, blue arrows indicate upregulation of the corresponding proteins or pathways; bidirectional arrows represent feedback interactions between the indicated pathways; red curved arrows represent inhibitory regulation. Created in BioRender. Kim, B. (2025) https://BioRender.com/r9g2odr (accessed on 28 October 2025).

**Table 1 cancers-17-03519-t001:** Redox-modulating mechanisms and translational roles of SH003 constituents.

Compound	Chelation Potential	Biomarker	Key Mechanism	Translational Relevance	Model	Reference
Baicalein (5,6,7-trihydroxy flavone core)	Possible Fe^2+^ binding via 5-OH/4-keto; weaker than catechol flavonoids	NRF2/ GPX4 redox balance	Promotes ROS-driven cytotoxicity; inhibits JAK2/STAT3 → GPX4 suppression → ferroptosis	Biomarker modulation (NRF2/GPX4); ferroptosis sensitizer	HCT116, DLD1, CRC xenograft	[67]
Chrysoeriol (3″-methoxy-4″,5,7-trihydroxy flavone)	Lacks catechol; Fe^2+^ binding via 5-OH/4-keto		Inhibits TNFα-induced EGR-1/CYP19; reduces estrogen biosynthesis	Supportive flavonoid; potential chemopreventive adjuvant in ER^+^ breast cancer	MCF-7	[68]
Cucurbitacin B/D (Triterpenoid)	No strong chelation; enhances ferroptosis via SLC7A11–GPX4 suppression and ROS accumulation	GPX4, SLC7A11	STAT3 inhibition (CuB strong; CuD possible); suppression of SLC7A11–GPX4	Ferroptosis biomarker (GPX4↓); immune modulation via STAT3 (CuB evidence, CuD inferred)	MCF-7, MDA-MB-231, CNE1 xenograft	[69,70]
Decursin (pyranocoumarin)	No catechol chelation; redox modulation via ER stress	CHOP	Induces ER stress (PERK–eIF2α–ATF4–CHOP); Nox4-mediated ROS; caspase-3/9 activation	Biomarker: CHOP↑; ER stress–apoptosis driver; potential chemo-combination (oxaliplatin)	HCT-116, HCT-8, MCF-7, MDA-MB-231, LNCaP, DU145, PC3, PANC-1, MiaPaCa-2	[71,72]
Formononetin (isoflavone)	No catechol chelation	NRF2, PI3K	Modulates KEAP1–NRF2 (GSTP1↑); PI3K/Akt–NRF2 linkage; ER stress (context-dependent)	Biomarker: NRF2 pathway; normal tissue protection; safe in oxaliplatin combinations	A549, HCT116, HT29, MCF-7, SH-SY5Y, CRC xenograft	[35,41,73]
Hispidulin (4″-methoxy-5,7-dihydroxy)	Limited Fe^2+^ binding (5,7-dihydroxy)	UPR/ CHOP	ROS-dependent apoptosis via mitochondrial dysfunction and ER stress (CHOP↑)	Translational apoptosis driver (UPR biomarker: CHOP↑, p-eIF2α↑)	HepG2, A549, NCI-H460	[74,75]
Luteolin (3″,4″-dihydroxy catechol moiety, 5,7-dihydroxy)	Strong Fe^2+^ chelation (catechol group)	GPX4, NRF2	Induces oxidative stress; downregulates NRF2–KEAP1–Cul3; triggers GPX4-dependent ferroptosis	Ferroptosis biomarkers (GPX4↓, lipid peroxidation, GSH/GSSG); immune activation (M1 polarization, CD8^+^ T-cell infiltration)	PC3, DU145; MC38, CT26; syngeneic mouse	[76,77]

Most mechanisms summarized here are supported by preclinical SH003 studies or constituent-derived evidence unless otherwise indicated. Arrows indicate direction of regulation: ↓ downregulation, ↑ upregulation. Abbreviations: Colorectal cancer (CRC), Cullin-3 (Cul3); Early growth response 1 (EGR-1); Estrogen receptor–positive (ER^+^); Reduced glutathione (GSH)/Oxidized glutathione (GSSG); Glutathione S-transferase Pi 1 (GSTP1); NAD(P)H quinone dehydrogenase 1 (NQO1); NADPH oxidase 4 (Nox4); Solute carrier family 7 member 11 (SLC7A11); Tumor necrosis factor alpha (TNFα).

**Table 2 cancers-17-03519-t002:** Key redox pathways targeted by SH003 and associated biomarkers with translational relevance.

Target	SH003 Modulation	Mechanistic Pathways	Biomarkers	Translational Relevance	Evidence Level	Refs:
Trx1/Trx2	TrxR inhibition (baicalein, reported); TRX2 speculative/unvalidated	↑ ASK1 activation, ↑ p53 stabilization, apoptosis/senescence	Trx1/Trx2 ratio, oxidation state	Established oxidative stress marker (clinical correlation); potential stratifier	Clinical correlation	[81]
GPX4	Inhibition by cucurbitacin D	↑ Ferroptotic cell death	GPX4 protein	Predictive marker for ferroptosis sensitivity (TNBC, NSCLC)	MDA-MB-231, Hs578T; A549, H460	[79]
NRF2	Suppression by baicalein, formononetin	↓ antioxidant genes, ↑ ROS	NRF2 localization, HO-1/NQO1	Context-dependent marker of adaptation and therapy resistance	HK-2, Wistar rat AKI	[87]
CHOP	Upregulation by decursin, hispidulin	↑ ER stress–mediated apoptosis	CHOP expression	Apoptosis stress marker (HCC, prostate, cervical models)	HepG2, PC-3, DU145, HeLa, PC-3 xenograft, HepG2 xenograft	[88]
TXNIP	Indirect upregulation	↑ ASK1 activation, oxidative apoptosis	TXNIP mRNA/ protein	Oxidative imbalance marker; therapeutic monitoring in nephrotoxicity	HK-2, Wistar rat	[89]

Evidence levels range from constituent inference to preclinical validation; only TRX1/TRX2 exhibit clinical correlation to date. Arrows indicate direction of regulation: ↓ downregulation, ↑ upregulation. Abbreviations: Apoptosis signal-regulating kinase 1 (ASK1); Cucurbitacin B (CuB); Cucurbitacin D (CuD); Docetaxel (DTX); Glycogen synthase kinase 3 beta (GSK3β); Heme oxygenase 1 (HO-1); Messenger RNA (mRNA); NAD(P)H quinone dehydrogenase 1 (NQO1); Tumor-associated macrophage (TAM); Thioredoxin reductase (TrxR); Thioredoxin 1/Thioredoxin 2 (Trx1/Trx2); Thioredoxin-interacting protein (TXNIP).

**Table 3 cancers-17-03519-t003:** Comparative clinical focus of natural products in breast and lung cancer and translational positioning of SH003.

Cancer Type	Main Clinical Focus of Natural Products	Example Interventions	SH003 Positioning
Breast cancer	Supportive care → QoL improvement, radiodermatitis prevention, perioperative anxiety, menopausal symptom relief, depression management	Broccoli sprout extract (isothiocyanates) [122], Polyphenol capsules (Curcumin, isoflavones, lignans) [123], Chicory root extract [124], chamomile gel [125], PureCyTonin (pollen extract) [126], Yokukansan (Kampo) [127], Shugan Jieyu San [128], calendula [129], Yiqi Yangyin decoction +chemo [130], curcumin [131].	Tumor-modifying phytomedicine → Mechanistic redox disruption & immune modulation [9].
Lung cancer	Disease-modifying strategies → Tumor suppression, drug resistance overcoming, immune modulation, ferroptosis induction	Oral decoctions after adjuvant chemo [132], Yiqi Qingdu prescription [133], multi-herbal combinations [134], BJIKT + ICIs [135], GQT *+* ICIs [136], Shenling Baizhu powder [137], LTTL [138].	Ferroptosis induction + NRF2 adaptation + immune remodeling → Translational bridge: preclinical evidence → Phase I feasibility [9].

Abbreviations: Bojungikki-tang (BJIKT); Chinese herbal medicine (CHM); Gegen Qinlian Tablets (GQT)*;* Immune checkpoint inhibitors (ICIs); Longteng Tongluo recipe (LTTL); Quality of life (QoL).

**Table 4 cancers-17-03519-t004:** Proposed exploratory trial design for SH003 combinations in NSCLC/TNBC.

Section	Item	Proposed Content
Population/Setting	Indications	EGFR-mutant NSCLC or TNBC with measurable, advanced disease; prior standard therapy permitted.
	Eligibility highlights	ECOG 0–1; adequate organ function; archival or fresh tumor tissue available for biomarker testing.
Arms (Phase Ib/II)	Arm A	Docetaxel + SH003.
	Arm B (control)	Docetaxel alone.
	Exploratory cohort	Anti-PD-1/PD-L1 + SH003 in PD-1–refractory subsets (signal-seeking).
Primary endpoints	Phase Ib	Dose-limiting toxicities (DLTs), MTD, RP2D (combination).
	Phase II	Progression-free survival (PFS; RECIST 1.1).
Key secondary endpoints	Efficacy & safety	ORR, DoR, DCR, OS; safety (AE/SAE, irAEs); patient-reported outcomes (QoL).
Stratification factors (pre-specified)	Genomic/biologic	KEAP1/NFE2L2 genotype; NRF2 target-gene signature (high vs. low); GPX4 expression (low vs. high); PD-L1 status.
	Clinical	Smoking history (to aid interpretation of metabolomic/circulating biomarker signals).
Correlative/Pharmacodynamic endpoints	Redox/ferroptosis	Tumor SLC7A11, GPX4; lipid peroxidation markers (4-HNE, MDA); NRF2 targets (NQO1, GCLC).
	ER-stress/mitochondria	CHOP (DDIT3); thioredoxin system (Trx/TrxR; including TRX2) as mitochondrial redox readouts.
	Immune remodeling	CD8^+^ infiltration; M1/M2 ratio; p-STAT3; PD-L1 (IHC).
	Circulating biomarkers	Serum SQSTM1 (p62)**;** exosome-based miRNA panel (e.g., miR-200c/21/210/96); cfDNA/ctDNA.
	Exosome metabolomics	Exploratory predictive biomarker collection in combination arms; functional immunologic causality to be tested prospectively.
Dosing/Safety	SH003	Start at safe monotherapy exposure from prior Phase I; stepwise escalation to combination RP2D (e.g., 2400–4800 mg/day), with close hepatic/hematologic monitoring.
	Docetaxel	Standard dose and schedule per label or institutional practice.
	Drug–drug considerations	Monitor for metabolism/transport interactions; pre-specify management of overlapping toxicities.
Interim/Statistics	Interim analyses	Planned futility/safety looks at pre-defined information fractions.
	Subgroup testing	Pre-specified interaction tests for KEAP1/NFE2L2 strata; estimate differential benefit in NRF2-hyperactivated tumors.

Abbreviations: Adverse event (AE); Circulating tumor DNA (ctDNA); Disease control rate (DCR); Duration of response (DoR); Dose-limiting toxicity (DLT); Immunohistochemistry (IHC); Immune-related adverse event (irAE); Malondialdehyde (MDA); Maximum tolerated dose (MTD); Objective response rate (ORR); Overall survival (OS); Progression-free survival (PFS); Quality of life (QoL); Recommended Phase 2 dose (RP2D).

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
