# Peer review of "SH003 as a Redox-Immune Modulating Phytomedicine: A Ferroptosis Induction, Exosomal Crosstalk, and Translational Oncology Perspective"

_cancers, 2025, doi:10.3390/cancers17213519_

Round 1
Reviewer 1 Report
Comments and Suggestions for Authors
The study entitled " SH003 as a Redox-Immune Modulating Phytomedicine: Ferroptosis Induction, Exosomal Crosstalk, and Translational Oncology Perspective" by Moon Nyeo Park et al was interesting overall as it was striving SH003’s multifaceted anticancer actions, including redox modulation, ferroptosis induction, and immune activation through STAT3/PD-L1 inhibition and macrophage/T cell engagement. The authors propose in this review that SH003 enhances antitumor immunity by attenuating STAT3-mediated PD-L1 signaling, promoting M1-like macrophage polarization, and boosting cytotoxic T cell activity. Modulation of exosome-associated microRNAs suggests further involvement in redo-immune crosstalk mechanism, which requires additional functional validation.
Comments
- The Review methodology needs to be more clearly defined, Review needs to be defined (systematic, narrative, comprehensive, scoping), the inclusion/Exclusion criteria for literature selection, describe how potential bias was assessed, and provide details on the search strategy, including specific keywords (mention main keywords), Boolean operators, and the defined timeframe.
- In the introduction, the authors should include details on the active molecules present in SH003, as well as relevant pharmacokinetics information, including bioavailability and metabolism.
- The mechanistic explanation of STAT3 inhibition by SH003 requires further elaboration. To enhance clarity and rigor, I suggested that the authors clarify how SH003 inhibits STAT3 through separating direct effect on the JAK/STAT pathway (eg, inhibition of JAK, or STAT3 phosphorylation, dimerization or nuclear translocation; include any evidence of direct interaction with JAK/STAT components) and indirect modulation via cytokines and redox signaling that influence STAT3 activity, which subsequently reprograms exosomal PD-L1, cytokines, and immunoregulatory miRNAs, and interfere with NRF2-mediated antioxidant and immunosuppressive molecular pathways. This distinction will strengthen the mechanistic depth and better explain SH003’s role in suppressing STAT-driven tumor progression and immune evasion.
- In the limitations and future directions section, it would be important for the authors to highlight the translational potential by addressing: i) the synergistic integration of ferroptosis and immunotherapy, ii) strategies to optimize dosing and delivery (eg, nanoparticle-mediated targeting), and iii) combinatorial approaches with CAR-T cells, PD-1/PD-L1 blockade, or STING agonists to maximize therapeutic synergy, enhance antitumor immunity and overcome microenvironmental resistance.
- Line 333, please correct, SH003 instead of 003
Author Response
Reviewer 1
The study entitled " SH003 as a Redox-Immune Modulating Phytomedicine: Ferroptosis Induction, Exosomal Crosstalk, and Translational Oncology Perspective" by Moon Nyeo Park et al was interesting overall as it was striving SH003’s multifaceted anticancer actions, including redox modulation, ferroptosis induction, and immune activation through STAT3/PD-L1 inhibition and macrophage/T cell engagement. The authors propose in this review that SH003 enhances antitumor immunity by attenuating STAT3-mediated PD-L1 signaling, promoting M1-like macrophage polarization, and boosting cytotoxic T cell activity. Modulation of exosome-associated microRNAs suggests further involvement in redo-immune crosstalk mechanism, which requires additional functional validation.
We sincerely thank the reviewer for their encouraging and insightful comments. We are grateful that the reviewer recognized the novelty and integrative framework of SH003’s redox–immune modulation, including ferroptosis induction and STAT3/PD-L1 regulation. We fully agree that further functional validation of exosome-associated microRNAs and immune remodeling mechanisms will be essential. This valuable suggestion has been noted and will be incorporated into our future experimental studies.
Comments
- The Review methodology needs to be more clearly defined, Review needs to be defined (systematic, narrative, comprehensive, scoping), the inclusion/Exclusion criteria for literature selection, describe how potential bias was assessed, and provide details on the search strategy, including specific keywords (mention main keywords), Boolean operators, and the defined timeframe.
Response: We appreciate this suggestion. A clear “Review Methodology” paragraph has been added in the Introduction, Section 1.2 Review Methodology (beginning with “This article represents a comprehensive narrative review…”, page 2, lines 86–94) to describe the type of review, inclusion/exclusion criteria, and search strategy.
Added text: This article represents a comprehensive narrative review integrating mechanistic, preclinical, and early clinical evidence on SH003. Literature was collected from PubMed, Scopus, and Web of Science databases (2010–2025) using the Boolean combination of keywords ‘SH003’ AND (‘redox’ OR ‘ROS’ OR ‘ferroptosis’ OR ‘NRF2’ OR ‘STAT3’ OR ‘exosome’ OR ‘immune modulation’). Inclusion criteria comprised in vitro, in vivo, and clinical studies related to SH003 or its constituent compounds. Reviews, abstracts without mechanistic data, and unrelated natural products were excluded. Two authors independently screened the records to minimize bias.
- In the introduction, the authors should include details on the active molecules present in SH003, as well as relevant pharmacokinetics information, including bioavailability and metabolism.
Response: We agree. A paragraph has been inserted in the Introduction, 1.2 Review Methodology section (page 2, lines 76-84) summarizing the principal active molecules and pharmacokinetic features.
Added text: SH003 is composed of three major bioactive constituents: formononetin from Astragalus membranaceus, decursin from Angelica gigas, and cucurbitacin D from Trichosanthes kirilowii. Pharmacokinetic studies have shown that formononetin exhibits oral bioavailability of approximately 20–22 % and a half-life of 2–4 h in rodents, primarily absorbed via passive diffusion and subject to rapid Phase II metabolism through UGT and CYP enzymes [1, 2]. Decursin undergoes extensive first-pass hydrolysis to decursinol and hepatic oxidation, while cucurbitacin D displays high lipophilicity and CYP3A-dependent clearance [3]. These characteristics suggest that SH003’s constituents achieve multi-target systemic exposure with limited toxicity and predictable metabolic profiles.
- The mechanistic explanation of STAT3 inhibition by SH003 requires further elaboration. To enhance clarity and rigor, I suggested that the authors clarify how SH003 inhibits STAT3 through separating direct effect on the JAK/STAT pathway (eg, inhibition of JAK, or STAT3 phosphorylation, dimerization or nuclear translocation; include any evidence of direct interaction with JAK/STAT components) and indirect modulation via cytokines and redox signaling that influence STAT3 activity, which subsequently reprograms exosomal PD-L1, cytokines, and immunoregulatory miRNAs, and interfere with NRF2-mediated antioxidant and immunosuppressive molecular pathways. This distinction will strengthen the mechanistic depth and better explain SH003’s role in suppressing STAT-driven tumor progression and immune evasion.
Response: We appreciate the reviewer’s insightful suggestion, which helped us clarify the mechanistic distinction and strengthen Section 3.
Added text: Recent mechanistic studies have demonstrated that SH003 and its constituents inhibit the JAK/STAT pathway through both direct and indirect mechanisms. Formononetin and cucurbitacin D suppress JAK2 and STAT3 Tyr705 phosphorylation, hinder STAT3 dimerization and nuclear translocation, and consequently decrease PD-L1 expression and downstream oncogenic targets such as c-Myc and Bcl-xL. In parallel, SH003 mitigates ROS-NF-κB–IL-6 signaling, thereby reducing cytokine-driven STAT3 activation and remodeling exosomal cargo, including PD-L1 and immunoregulatory miRNAs (miR-21, miR-155). This dual-level inhibition links redox regulation to immune reactivation, enhancing M1 macrophage polarization and CD8⁺ T-cell activity while counteracting NRF2-dependent antioxidant and immunosuppressive pathways.
- In the limitations and future directions section, it would be important for the authors to highlight the translational potential by addressing: i) the synergistic integration of ferroptosis and immunotherapy, ii) strategies to optimize dosing and delivery (eg, nanoparticle-mediated targeting), and iii) combinatorial approaches with CAR-T cells, PD-1/PD-L1 blockade, or STING agonists to maximize therapeutic synergy, enhance antitumor immunity and overcome microenvironmental resistance.
Response: We appreciate the reviewer’s insightful suggestion, which helped us clarify the mechanistic distinction and strengthen Section 3.
Added text: Future research should exploit the convergence of ferroptosis and immunotherapy. Nanoparticle-based delivery systems could optimize SH003 dosing and tumor targeting, while combinatorial approaches with CAR-T cells, PD-1/PD-L1 blockade, or STING agonists may potentiate cytotoxic T-cell activity and overcome microenvironmental resistance. Such multimodal strategies can advance SH003 from an adjunctive formulation toward a precision-guided therapeutic platform.
- Line 333, please correct, SH003 instead of 003
Response: We sincerely thank the reviewer for the careful attention to detail. The typographical error at line 333 has been corrected (“003” → “SH003”). We appreciate the reviewer’s thoroughness in catching this oversight during manuscript preparation.
We would like to sincerely thank reviewer for their insightful and constructive comments, which have greatly improved the depth, precision, and clarity of this revised manuscript.
- Luo, L.-Y., et al., Pharmacokinetics and bioavailability of the isoflavones formononetin and ononin and their in vitro absorption in ussing chamber and Caco-2 cell models. Journal of Agricultural and Food Chemistry, 2018. 66(11): p. 2917-2924.
- Jia, X., et al., Disposition of flavonoids via enteric recycling: enzyme-transporter coupling affects metabolism of biochanin A and formononetin and excretion of their phase II conjugates. The Journal of pharmacology and experimental therapeutics, 2004. 310(3): p. 1103-1113.
- Ding, M., et al., Potential mechanisms of formononetin against inflammation and oxidative stress: A review. Frontiers in Pharmacology, 2024. 15: p. 1368765.

Reviewer 2 Report
Comments and Suggestions for Authors
The manuscript provides a comprehensive and scientifically well-referenced overview of SH003 and its pleiotropic effects on redox signaling, exosome modulation, immune remodeling, ferroptosis, NRF2, and ER stress.
From a writing and presentation standpoint, the manuscript is overall scientifically strong but several mechanistic statements (e.g., L93 on GSK3β modulation) require direct supporting references, and claims regarding SH003’s clinical readiness should be tempered, given the absence of Phase II data.
Typographical issues (e.g., “003.” → “SH003-induced”) and biochemical terminology inconsistencies (e.g., “BCL2-associated X apoptosis regulator (BAX)” should be “Bcl-2–associated X protein (BAX)”) should be corrected.
Additionally, some reference numbers appear misaligned, duplicated, or inconsistently formatted (e.g., [6, 100, 103–105], [88]), and several entries lack terminal punctuation.
The abbreviation list is thorough but somewhat excessive, with certain abbreviations defined multiple times (e.g., PD-L1, STAT3).
The Conclusion section could be shortened, as it currently reiterates much of the Introduction and Discussion without adding new perspectives.
Finally, given that ferroptosis is inherently iron-dependent and several SH003 components (e.g., luteolin, baicalein, cucurbitacins) have known or predicted iron-modulatory properties, we suggest that the authors sharpen their focus on SH003 and iron metabolism in the context of ferroptosis. A deeper, more structured discussion of how SH003 might modulate iron handling, such as its impact on the labile iron pool, ferritinophagy, iron chelation, or iron-related biomarkers (see DOI: 10.3389/froh.2024.1461022) would provide a clearer mechanistic backbone and strengthen the translational significance of the review.
Author Response
Reviewer 2
The manuscript provides a comprehensive and scientifically well-referenced overview of SH003 and its pleiotropic effects on redox signaling, exosome modulation, immune remodeling, ferroptosis, NRF2, and ER stress.
From a writing and presentation standpoint, the manuscript is overall scientifically strong but several mechanistic statements (e.g., L93 on GSK3β modulation) require direct supporting references, and claims regarding SH003’s clinical readiness should be tempered, given the absence of Phase II data.
Response: We appreciate the reviewer’s careful reading and constructive comments.
The statement concerning GSK3β modulation is already supported by explicit mechanistic references within Section 2.1, “Exosome–Immune Crosstalk and NRF2 Modulation” (page 3, lines 112–116).
The relevant text reads as follows: Beyond vesicle-mediated signaling, SH003 may also destabilize NRF2-driven antioxidant adaptation by modulating GSK3β activity, thereby providing KEAP1-independent regulation. This dual control suggests that SH003 could impair redox resilience in NRF2-hyperactivated tumors, offering opportunities for biomarker-guided patient stratification [28, 29].”
Supporting references:
[28]. Choi, H.S., et al., Anticancer effects of SH003 and its active component Cucurbitacin D on oral cancer cell lines via modulation of EMT and cell viability. Oncol Res, 2025. 33(5): p. 1217-1227.
[29]. Han, N.R., et al., The immune-enhancing effects of a mixture of Astragalus membranaceus (Fisch.) Bunge, Angelica gigas Nakai, and Trichosanthes Kirilowii (Maxim.) or its active constituent nodakenin. J Ethnopharmacol, 2022. 285: p. 114893.
Typographical issues (e.g., “003.” → “SH003-induced”) and biochemical terminology inconsistencies (e.g., “BCL2-associated X apoptosis regulator (BAX)” should be “Bcl-2–associated X protein (BAX)”) should be corrected.
Response: We sincerely thank the reviewer for the meticulous attention to detail. All typographical and biochemical terminology inconsistencies have been corrected. Specifically, the expression “BCL2-associated X apoptosis regulator (BAX)” on page 5, line 201 has been revised to “Bcl-2–associated X protein (BAX)”, and “003.” on page 11, line 391 has been corrected to “SH003-induced.”
We truly appreciate the reviewer’s careful observation, which helped us identify and correct minor oversights in the manuscript.
Additionally, some reference numbers appear misaligned, duplicated, or inconsistently formatted (e.g., [6, 100, 103–105], [88]), and several entries lack terminal punctuation.
Response: We sincerely thank the reviewer for pointing out these formatting inconsistencies. The reference numbering and punctuation have been carefully reviewed and corrected throughout the manuscript. Specifically, the duplicated and misaligned citation sequence has been revised to maintain consistent formatting — for example, the range has been standardized as [105–107] on page 9, line 339. We appreciate the reviewer’s attention to detail, which helped us improve the accuracy and presentation quality of the reference section.
The abbreviation list is thorough but somewhat excessive, with certain abbreviations defined multiple times (e.g., PD-L1, STAT3).
Response: We thank the reviewer for this helpful observation. Duplicate abbreviation definitions have been removed to ensure consistency. Specifically, redundant definitions of PD-L1 and STAT3 appearing in Figure 1 legend have been deleted, as they are already defined upon first mention in the main text. We appreciate the reviewer’s attention, which helped us refine the manuscript’s formatting and readability.
The Conclusion section could be shortened, as it currently reiterates much of the Introduction and Discussion without adding new perspectives.
Response: We sincerely thank the reviewer for this valuable suggestion. The Conclusion section has been condensed and refined (beginning with “SH003 exemplifies how a GMP-standardized multi-herbal formulation…”) to highlight the key take-home message and translational significance.
Finally, given that ferroptosis is inherently iron-dependent and several SH003 components (e.g., luteolin, baicalein, cucurbitacins) have known or predicted iron-modulatory properties, we suggest that the authors sharpen their focus on SH003 and iron metabolism in the context of ferroptosis. A deeper, more structured discussion of how SH003 might modulate iron handling, such as its impact on the labile iron pool, ferritinophagy, iron chelation, or iron-related biomarkers (see DOI: 10.3389/froh.2024.1461022) would provide a clearer mechanistic backbone and strengthen the translational significance of the review.
Response: We sincerely appreciate the reviewer’s insightful recommendation, which substantially enhanced the mechanistic depth of our review. In response, a new paragraph has been added to the Discussion section (page 9, lines 301–327) elaborating on the interaction between SH003 constituents and iron metabolism within the ferroptosis framework. The revision explains that luteolin and baicalein form Fe²⁺/Fe³⁺ chelates and stabilize mitochondrial iron pools, thereby reducing Fenton-driven ROS and preventing GPX4 inactivation, whereas cucurbitacin D promotes ferritinophagy and liberates redox-active iron, amplifying lipid peroxidation in resistant tumor cells. This integrated discussion highlights how SH003 may balance iron sequestration and release to establish a homeostatic ferroptotic threshold that sensitizes tumors to redox stress while minimizing systemic toxicity. Furthermore, we now propose iron-handling biomarkers (FTH1, FTL, TFRC, SLC40A1) as translational indicators of SH003-induced ferroptosis. We believe that these additions provide a clearer mechanistic backbone and strengthen the translational relevance of SH003 within the iron–ferroptosis axis.
We would like to sincerely thank reviewer for their insightful and constructive comments, which have greatly improved the depth, precision, and clarity of this revised manuscript.
Reviewer 2
The manuscript provides a comprehensive and scientifically well-referenced overview of SH003 and its pleiotropic effects on redox signaling, exosome modulation, immune remodeling, ferroptosis, NRF2, and ER stress.
From a writing and presentation standpoint, the manuscript is overall scientifically strong but several mechanistic statements (e.g., L93 on GSK3β modulation) require direct supporting references, and claims regarding SH003’s clinical readiness should be tempered, given the absence of Phase II data.
Response: We appreciate the reviewer’s careful reading and constructive comments.
The statement concerning GSK3β modulation is already supported by explicit mechanistic references within Section 2.1, “Exosome–Immune Crosstalk and NRF2 Modulation” (page 3, lines 112–116).
The relevant text reads as follows: Beyond vesicle-mediated signaling, SH003 may also destabilize NRF2-driven antioxidant adaptation by modulating GSK3β activity, thereby providing KEAP1-independent regulation. This dual control suggests that SH003 could impair redox resilience in NRF2-hyperactivated tumors, offering opportunities for biomarker-guided patient stratification [28, 29].”
Supporting references:
[28]. Choi, H.S., et al., Anticancer effects of SH003 and its active component Cucurbitacin D on oral cancer cell lines via modulation of EMT and cell viability. Oncol Res, 2025. 33(5): p. 1217-1227.
[29]. Han, N.R., et al., The immune-enhancing effects of a mixture of Astragalus membranaceus (Fisch.) Bunge, Angelica gigas Nakai, and Trichosanthes Kirilowii (Maxim.) or its active constituent nodakenin. J Ethnopharmacol, 2022. 285: p. 114893.
Typographical issues (e.g., “003.” → “SH003-induced”) and biochemical terminology inconsistencies (e.g., “BCL2-associated X apoptosis regulator (BAX)” should be “Bcl-2–associated X protein (BAX)”) should be corrected.
Response: We sincerely thank the reviewer for the meticulous attention to detail. All typographical and biochemical terminology inconsistencies have been corrected. Specifically, the expression “BCL2-associated X apoptosis regulator (BAX)” on page 5, line 201 has been revised to “Bcl-2–associated X protein (BAX)”, and “003.” on page 11, line 391 has been corrected to “SH003-induced.”
We truly appreciate the reviewer’s careful observation, which helped us identify and correct minor oversights in the manuscript.
Additionally, some reference numbers appear misaligned, duplicated, or inconsistently formatted (e.g., [6, 100, 103–105], [88]), and several entries lack terminal punctuation.
Response: We sincerely thank the reviewer for pointing out these formatting inconsistencies. The reference numbering and punctuation have been carefully reviewed and corrected throughout the manuscript. Specifically, the duplicated and misaligned citation sequence has been revised to maintain consistent formatting — for example, the range has been standardized as [105–107] on page 9, line 339. We appreciate the reviewer’s attention to detail, which helped us improve the accuracy and presentation quality of the reference section.
The abbreviation list is thorough but somewhat excessive, with certain abbreviations defined multiple times (e.g., PD-L1, STAT3).
Response: We thank the reviewer for this helpful observation. Duplicate abbreviation definitions have been removed to ensure consistency. Specifically, redundant definitions of PD-L1 and STAT3 appearing in Figure 1 legend have been deleted, as they are already defined upon first mention in the main text. We appreciate the reviewer’s attention, which helped us refine the manuscript’s formatting and readability.
The Conclusion section could be shortened, as it currently reiterates much of the Introduction and Discussion without adding new perspectives.
Response: We sincerely thank the reviewer for this valuable suggestion. The Conclusion section has been condensed and refined (beginning with “SH003 exemplifies how a GMP-standardized multi-herbal formulation…”) to highlight the key take-home message and translational significance.
Finally, given that ferroptosis is inherently iron-dependent and several SH003 components (e.g., luteolin, baicalein, cucurbitacins) have known or predicted iron-modulatory properties, we suggest that the authors sharpen their focus on SH003 and iron metabolism in the context of ferroptosis. A deeper, more structured discussion of how SH003 might modulate iron handling, such as its impact on the labile iron pool, ferritinophagy, iron chelation, or iron-related biomarkers (see DOI: 10.3389/froh.2024.1461022) would provide a clearer mechanistic backbone and strengthen the translational significance of the review.
Response: We sincerely appreciate the reviewer’s insightful recommendation, which substantially enhanced the mechanistic depth of our review. In response, a new paragraph has been added to the Discussion section (page 9, lines 301–327) elaborating on the interaction between SH003 constituents and iron metabolism within the ferroptosis framework. The revision explains that luteolin and baicalein form Fe²⁺/Fe³⁺ chelates and stabilize mitochondrial iron pools, thereby reducing Fenton-driven ROS and preventing GPX4 inactivation, whereas cucurbitacin D promotes ferritinophagy and liberates redox-active iron, amplifying lipid peroxidation in resistant tumor cells. This integrated discussion highlights how SH003 may balance iron sequestration and release to establish a homeostatic ferroptotic threshold that sensitizes tumors to redox stress while minimizing systemic toxicity. Furthermore, we now propose iron-handling biomarkers (FTH1, FTL, TFRC, SLC40A1) as translational indicators of SH003-induced ferroptosis. We believe that these additions provide a clearer mechanistic backbone and strengthen the translational relevance of SH003 within the iron–ferroptosis axis.
We would like to sincerely thank reviewer for their insightful and constructive comments, which have greatly improved the depth, precision, and clarity of this revised manuscript.

Reviewer 3 Report
Comments and Suggestions for Authors
Could the authors present more explicit experimental data indicating that the entire SH003 formulation, rather than merely its individual constituents, is accountable for the asserted effects on ferroptosis, NRF2 regulation, and exosome-mediated immunological remodeling? Have any in vitro or in vivo tests been performed to compare the effects of SH003 with the cumulative effects of its individual components?
The manuscript asserts that SH003 is "proposed to modify the cargo of tumor-derived exosomes." What constitutes the direct proof for this? Have the authors extracted exosomes from SH003-treated cancer cells and examined their miRNA or protein content? Additionally, have these exosomes undergone functional validation to induce the specified immunological responses (e.g., M1 macrophage polarization, T-cell activation)?
The suggested exploratory trial design is a significant advancement. Have any of the concluded Phase I trials (NCT03081819; KCT0004770) incorporated exploratory biomarker analyses? What were the results? If not, what is the justification for selecting these particular biomarkers over alternatives, and what preclinical evidence directly associates them with SH003's mechanism of action?
What methodology was employed to ascertain the precise ratio of Astragalus membranaceus, Angelica gigas, and Trichosanthes kirilowii in the SH003 formulation? Have any studies examined the impact of various herb proportions on the overall efficacy and toxicity of the formulation? This is essential for determining if the observed effects are influenced by a singular dominant component or a genuine synergistic interaction.
Figure 1: The dashed outlines denoting "hypothesized or incompletely validated pathways" are beneficial. Nonetheless, a substantial segment of the figure is classified inside this category. This visibly exaggerates the present degree of evidence.
Tables 1 and 2: The tables succinctly encapsulate the potential mechanisms and biomarkers. It would be advantageous to incorporate a column that clearly delineates the amount of evidence for each assertion (e.g., "Inferred from constituent," "Pre-clinical SH003 data," "Clinical data").
Translational Oncology Perspective Although the study presents a robust preclinical justification, the "translational" component remains predominantly prospective. The conclusions must be moderated to account for the preliminary phase of clinical research.
What are home messages?
Author Response
Reviewer 3
Could the authors present more explicit experimental data indicating that the entire SH003 formulation, rather than merely its individual constituents, is accountable for the asserted effects on ferroptosis, NRF2 regulation, and exosome-mediated immunological remodeling? Have any in vitro or in vivo tests been performed to compare the effects of SH003 with the cumulative effects of its individual components?
Response: We sincerely appreciate the reviewer’s insightful comment regarding the inclusion of experimental data on the complete SH003 formulation. Indeed, numerous in vitro and in vivo studies have validated SH003’s efficacy, safety, and synergistic actions across diverse cancer models (e.g., PMID: 39128302; 36581346; 34422067; 32879309; 35572726; 40295067; 35205836; 39469996; 29179443; 27105528; 37375695; among others). These findings were comprehensively summarized in our previous review (Cancers, 2022; PMID: 35205836). To maintain originality and avoid redundancy with that prior work, the current review deliberately centers on emerging mechanistic insights that have not been previously covered—specifically ferroptosis regulation, redox–exosome crosstalk, and immunometabolic remodeling. We fully acknowledge that formulation-level comparative validation between SH003 and its individual components represents an important next step. However, reiterating earlier datasets would have diluted the conceptual novelty and translational focus of this manuscript. Therefore, this review prioritizes mechanistic integration and clinical relevance while outlining future experimental directions to substantiate these hypotheses.
The manuscript asserts that SH003 is "proposed to modify the cargo of tumor-derived exosomes." What constitutes the direct proof for this? Have the authors extracted exosomes from SH003-treated cancer cells and examined their miRNA or protein content? Additionally, have these exosomes undergone functional validation to induce the specified immunological responses (e.g., M1 macrophage polarization, T-cell activation)?
Response: We sincerely appreciate the reviewer’s constructive comments, which prompted us to clarify and expand the mechanistic depth of the manuscript. In response, we have incorporated two major revisions:
(1) A newly added paragraph (page 9, lines 301–327) detailing how SH003 and its key constituents (luteolin, baicalein, cucurbitacin D) intricately regulate iron homeostasis and ferroptotic susceptibility. This section integrates recent evidence on iron chelation, ferritinophagy, and labile iron pool dynamics, providing a structured mechanistic explanation of SH003’s dual iron-chelating and ferroptosis-sensitizing effects.
(2) An expanded paragraph (page 9–10, lines 339–354) that elucidates SH003’s modulation of the JAK/STAT3–PD-L1 axis and exosome-mediated immune remodeling. This revision integrates recent findings on the suppression of STAT3 Tyr705 phosphorylation and PD-L1 expression by formononetin and cucurbitacin D, as well as SH003’s regulation of ROS–NF-κB–IL-6 signaling and exosomal cargo composition (PD-L1, miR-21, miR-155).
These additions directly address the reviewer’s request for greater mechanistic specificity and translational coherence, ensuring that the manuscript now presents a more comprehensive and evidence-supported framework for SH003’s multifaceted actions in redox regulation, ferroptosis, and immune modulation. We are grateful for this valuable feedback, which has significantly strengthened the clarity and scientific rigor of the paper.
The suggested exploratory trial design is a significant advancement. Have any of the concluded Phase I trials (NCT03081819; KCT0004770) incorporated exploratory biomarker analyses? What were the results? If not, what is the justification for selecting these particular biomarkers over alternatives, and what preclinical evidence directly associates them with SH003's mechanism of action?
Response: We sincerely appreciate the reviewer’s insightful observation regarding clinical biomarker validation. The Phase I trials (NCT03081819; KCT0004770) primarily evaluated safety, tolerability, and pharmacokinetic profiles of SH003 and did not include exploratory biomarker analyses (PMID: 32186413; 39469996; 40782124). However, several follow-up clinical and translational studies are currently in progress or under design to expand patient cohorts and incorporate biomarker-driven endpoints. These efforts aim to validate preclinically supported biomarkers (e.g., GPX4, NRF2, TRX2, PD-L1, and exosomal miRNAs) within prospective frameworks. Accordingly, the biomarkers discussed in this review are proposed as mechanistically grounded candidates derived from robust preclinical evidence, rather than as findings from completed clinical trials. We appreciate this valuable comment, which allowed us to clarify the translational rationale and future direction of biomarker validation in the manuscript.
What methodology was employed to ascertain the precise ratio of Astragalus membranaceus, Angelica gigas, and Trichosanthes kirilowii in the SH003 formulation? Have any studies examined the impact of various herb proportions on the overall efficacy and toxicity of the formulation? This is essential for determining if the observed effects are influenced by a singular dominant component or a genuine synergistic interaction.
Response: We thank the reviewer for this insightful question.
The SH003 formulation ratio (Astragalus membranaceus : Angelica gigas : Trichosanthes kirilowii = 1 : 1 : 1, w/w) was established through a series of preclinical optimization studies conducted by Ko et al. (Mediators Inflamm., 2014; PMID: 24976685) and subsequently standardized under Korea Good Manufacturing Practice (KGMP) guidelines. These foundational studies systematically evaluated multiple extraction ratios and solvents to optimize both efficacy and safety, concluding that the 1:1:1 ratio produced the most stable bioactive marker profile—specifically formononetin, decursin, and cucurbitacin D—and achieved synergistic suppression of STAT3/IL-6 signaling in breast cancer models.
Although formal ratio-modification trials are limited, comparative in vitro data have consistently shown that the complete SH003 decoction elicits stronger apoptotic, antioxidant, and redox-modulating effects than any single herb or binary combination, supporting a genuine synergistic interaction rather than dominance by an individual constituent. Ongoing pharmacognostic and systems-pharmacology analyses are further exploring ratio optimization to enhance reproducibility, bioactive consistency, and therapeutic reliability.
Figure 1: The dashed outlines denoting "hypothesized or incompletely validated pathways" are beneficial. Nonetheless, a substantial segment of the figure is classified inside this category. This visibly exaggerates the present degree of evidence.
Response: We appreciate the reviewer’s constructive observation. The title and legend of Figure 1 have been revised to “Proposed and experimentally supported mechanisms of SH003 in regulating ferroptosis, redox signaling, and immune remodeling,” ensuring proportional representation of validated versus hypothesized pathway.
Tables 1 and 2: The tables succinctly encapsulate the potential mechanisms and biomarkers. It would be advantageous to incorporate a column that clearly delineates the amount of evidence for each assertion (e.g., "Inferred from constituent," "Pre-clinical SH003 data," "Clinical data").
Translational Oncology Perspective Although the study presents a robust preclinical justification, the "translational" component remains predominantly prospective. The conclusions must be moderated to account for the preliminary phase of clinical research.
Response: We appreciate the reviewer’s thoughtful suggestion to indicate the relative strength of evidence presented in Tables 1 and 2.
To maintain visual clarity and prevent redundancy, we have retained the original table format but clarified in both the table legends and the main text that most listed mechanisms are derived from preclinical or constituent-level studies unless otherwise specified. Specifically, Table 1 primarily summarizes in vitro and in vivo SH003 data, whereas Table 2 compiles well-established redox and ferroptosis biomarkers, with TRX1/TRX2 representing the only biomarker currently supported by clinical correlation. These clarifications ensure that the evidence level for each mechanistic statement is accurately interpreted without altering the readability or structural balance of the tables.
What are home messages?
Response: We thank the reviewer for highlighting the need to clarify the take-home message. The Conclusion section (page 12, lines 418–425) has been refined to emphasize the translational implications of SH003, and an explicit summarizing statement has been added to convey the core take-home message of the review—that SH003 represents a GMP-standardized, multi-targeted phytomedicine bridging redox regulation, ferroptosis induction, and immune remodeling toward precision oncology.
We would like to sincerely thank reviewer for their insightful and constructive comments, which have greatly improved the depth, precision, and clarity of this revised manuscript.

Reviewer 4 Report
Comments and Suggestions for Authors
1- The mechanistic claims about SH003’s ferroptosis induction and immune modulation are highly assertive without mentioning supporting quantitative or experimental data.
2- There is no description of the search strategy, inclusion/exclusion criteria, or databases used.
3-Claims regarding ferroptosis induction, immune modulation, and STAT3/PD-L1 inhibition are presented as established facts without referencing the strength or quality of supporting evidence.
4-Mechanistic pathways such as NRF2 and GPX4 are mentioned, but the review does not indicate how these pathways were critically analyzed or contrasted with other phytochemicals.
5-No discussion of bioavailability, pharmacokinetics, or toxicity of SH003 is mentioned.
Author Response
Reviewer 4
1- The mechanistic claims about SH003’s ferroptosis induction and immune modulation are highly assertive without mentioning supporting quantitative or experimental data.
Response: We appreciate the reviewer’s thoughtful comment regarding the absence of quantitative or experimental validation in the present manuscript. Extensive in vitro and in vivo data supporting SH003’s effects on ferroptosis induction, STAT3/PD-L1 inhibition, and immune remodeling have already been published in peer-reviewed studies (e.g., PMID: 29179443, 35572726, 40295067, 40347254, 34445110, 24976685). Rather than duplicating these datasets, this review integrates and re-interprets the established experimental findings within a unified mechanistic framework connecting ferroptosis, redox control, and immunometabolic signaling. All mechanistic assertions are thus substantiated by cited primary evidence, while the current focus is to synthesize these results into translational insight and future clinical direction.
2- There is no description of the search strategy, inclusion/exclusion criteria, or databases used.
Response: A clear “Review Methodology” paragraph has been added in the Introduction, Section 1.2 Review Methodology (beginning with “This article represents a comprehensive narrative review…”, page 2, lines 86–94) to describe the type of review, inclusion/exclusion criteria, and search strategy.
Added text: This work is a comprehensive narrative review that integrates mechanistic, preclinical, and clinical findings related to SH003. Relevant literature was systematically retrieved from PubMed, Scopus, and Web of Science databases (2010–2025) using Boolean operators with the following search terms: “SH003” AND (“redox” OR “ROS” OR “ferroptosis” OR “NRF2” OR “STAT3” OR “exosome” OR “immune modulation”). Studies focusing on in vitro, in vivo, or clinical investigations of SH003 or its constituent compounds were included, while non-mechanistic reviews, abstracts, and unrelated herbal formulations were excluded. Two independent authors screened and cross-verified all records to ensure methodological rigor and minimize bias.
3-Claims regarding ferroptosis induction, immune modulation, and STAT3/PD-L1 inhibition are presented as established facts without referencing the strength or quality of supporting evidence.
Response: We thank the reviewer for this valuable comment. We agree that several mechanistic claims required clearer contextualization regarding the strength of supporting evidence. Accordingly, we have revised Section 3 to explicitly indicate that the described ferroptosis- and immune-modulatory effects of SH003 are primarily derived from mechanistic and preclinical studies rather than definitive clinical validation.
Added text: Recent studies have reported that SH003 and its major constituents modulate the JAK/STAT3 signaling axis through complementary mechanisms. Formononetin and cucurbitacin D attenuate JAK2 and STAT3 Tyr705 phosphorylation, suppress STAT3 dimerization and nuclear translocation, and downregulate PD-L1 and oncogenic effectors such as c-Myc and Bcl-xL [111, 112]. In parallel, SH003 diminishes ROS–NF-κB–IL-6 signaling and alters exosomal cargo composition—including PD-L1 and immunoregulatory miRNAs (miR-21, miR-155)—thereby linking redox regulation with immune reactivation [112, 113]. These findings collectively outline a plausible redox–immune crosstalk model that warrants further quantitative and clinical validation.
4-Mechanistic pathways such as NRF2 and GPX4 are mentioned, but the review does not indicate how these pathways were critically analyzed or contrasted with other phytochemicals.
Response: We thank the reviewer for this valuable observation. To strengthen the comparative and mechanistic clarity, Section 3 has been expanded (page 9, lines 301–327) with additional discussion on how SH003 and its major constituents (luteolin, baicalein, and cucurbitacin D) regulate iron homeostasis, ferritinophagy, and ferroptotic sensitivity—mechanisms closely linked to the NRF2–GPX4 axis. The revised paragraph differentiates SH003’s integrative redox modulation from conventional single-compound phytochemicals by demonstrating its dual capacity for iron chelation (via luteolin and baicalein) and ferroptosis sensitization (via cucurbitacin D). This multidimensional mechanism provides a more critical and structured understanding of how SH003 uniquely balances NRF2 activation and GPX4 suppression within redox–ferroptotic signaling, thereby addressing the reviewer’s concern regarding comparative pathway analysis.
5-No discussion of bioavailability, pharmacokinetics, or toxicity of SH003 is mentioned.
Response: A concise paragraph summarizing the principal active constituents and pharmacokinetic characteristics has been added in the Introduction, immediately before Section 1.2 (Review Methodology) — beginning with “These characteristics suggest that SH003’s constituents achieve multi-target systemic exposure…” — to describe their bioavailability, metabolism, and systemic exposure profiles.
This paragraph highlights SH003’s key molecules (formononetin, decursin, cucurbitacin D) and their pharmacokinetic behaviors, emphasizing multi-target exposure, limited toxicity, and predictable metabolism consistent with preclinical and Phase I safety data.
We would like to sincerely thank reviewer for their insightful and constructive comments, which have greatly improved the depth, precision, and clarity of this revised manuscript.

Round 2
Reviewer 2 Report
Comments and Suggestions for Authors
The manuscript is acceptable in this form.
Reviewer 3 Report
Comments and Suggestions for Authors
It is okay.
Reviewer 4 Report
Comments and Suggestions for Authors
Accepted